# Effects of Pickleball Intervention on the Self-Esteem and Symptoms of Patients with Schizophrenia

**DOI:** 10.3390/sports13010021

**Published:** 2025-01-14

**Authors:** Tsai-Chieh Chien, Chao-Chien Chen

**Affiliations:** 1Department of Occupational Therapy, Asia University, Taichung 41354, Taiwan; ball0818@asia.edu.tw; 2Department of Leisure and Recreation Management, Asia University, Taichung 41354, Taiwan

**Keywords:** schizophrenia, pickleball, self-esteem, mental health

## Abstract

*Background:* Schizophrenia is classified by the World Health Organization (WHO) as one of the top ten diseases contributing to the global medical economic burden. Some studies have pointed out that exercise is effective for physical and mental health, as well as cognition. We hypothesized that participation in pickleball intervention would lead to improved self-esteem and reduced psychiatric symptoms in schizophrenia patients. *Method:* We recruited participants with schizophrenia from a long-term care facility and a regional hospital, dividing them into two groups based on the institutions. The experimental group underwent a nine-week pickleball exercise intervention, with sessions three times a week, each lasting 90 min, and a Dink ball test was conducted weekly. Questionnaires on self-esteem and the short-form health survey were collected both before and after the intervention. *Result:* This trial included 30 patients, divided equally into the experimental group (n = 15) and the control group (n = 15). After the nine-week pickleball intervention, there were no significant differences between the experimental and control groups in the Rosenberg Self-Esteem Scale (RSES) (*p* = 0.153) or the Brief Symptom Rating Scale (BSRS-5) (*p* = 0.289). However, the Dink test scores in the experimental group showed significant improvements in physical activity capabilities and attention over time, with average hit counts increasing from 5.3 ± 1.5 to 10.7 ± 2.3 (*p* < 0.01). *Conclusions:* Although the pickleball intervention did not yield significant differences in self-esteem and symptom measures between groups, the improvements observed in physical performance and attention in the experimental group suggest that exercise remains a feasible complementary approach for managing schizophrenia symptoms. Further research with larger sample sizes is recommended.

## 1. Introduction

Statistics from the World Health Organization (WHO) indicate that around 15% of individuals aged 60 and above are affected by mental health issues, including depression, anxiety, and dementia. These conditions constitute significant health concerns that impact the older demographic significantly [1]. Schizophrenia comprises a spectrum of complex mental health disorders with an unclear cause. Those affected experience pronounced disruptions in cognition, thought processes, and behavior [2], and it has a global lifetime occurrence rate of about 1% [3]. Compelling evidence supports the notion that consistent physical exercise improves mental health and is associated with better overall and social functioning in individuals with schizophrenia [4,5]. An increasing body of research indicates that physical exercise provides significant neurobiological and psychological benefits for individuals with mental illnesses. Physical activities can promote positive changes in brain structure, particularly by increasing hippocampal volume, which further enhances cognitive functions related to memory [6]. Additionally, exercise can reduce cortisol levels, thereby alleviating stress and improving mood [7]. This effect is particularly notable in addressing emotional dysregulation, such as anxiety and depressive symptoms, which are commonly observed in individuals with mental illnesses [8]. Furthermore, regular physical exercise contributes to improvements in attention and executive functioning, which enhance the functional performance of daily living activities [9]. Therefore, physical exercise represents a promising non-pharmacological intervention for addressing neurobiological and psychological challenges in individuals with mental disorders. Additionally, higher levels of physical activity are consistently linked with reduced instances of insomnia, enhanced cognitive and social functioning, and greater satisfaction with life when contrasted with lower activity levels [10]. Nevertheless, individuals suffering from diverse mental illnesses, including schizophrenia, are often at an elevated risk of developing additional health issues due to factors such as obesity, dyslipidemia, smoking habits, hypertension, hyperglycemia, and a sedentary lifestyle, all of which can further degrade their mental, physical, and social well-being [11,12]. Self-esteem, defined as an individual’s overall evaluation of their worth, plays a critical role in psychological well-being [13]. Research has shown that individuals with schizophrenia often experience low self-esteem, which is linked to negative symptoms and impaired social functioning [14]. Physical activity, particularly in group settings, has been identified as a potential intervention to improve self-esteem through enhanced physical competence, self-efficacy, and social connectedness [15]. This study aims to explore the effects of pickleball, a novel group-based sport, on self-esteem and psychiatric symptoms in individuals with schizophrenia, addressing the gap in the literature by employing an intervention-focused design. Pickleball is an emerging sport that combines elements of tennis, badminton, and table tennis. It is characterized by its simplicity, moderate intensity, and slower pace, making it particularly suitable for individuals with limited coordination or less experience in physical activities, including those with schizophrenia. Studies have shown that pickleball helps improve muscle strength, balance, and reaction time, while also enhancing mental health by reducing stress and fostering social interaction. Research indicates that team sports enhance participants’ social connections and self-confidence. As a team-based sport, pickleball involves doubles play and competitive activities, which promote interaction and mutual support among participants, particularly benefiting individuals with schizophrenia by mitigating social withdrawal. Furthermore, pickleball has significant effects on improving reaction time and attention, contributing to enhanced daily functioning and focus for patients [16,17,18]. Various sports programs, like those evaluated by Walter et al., have been found to offer short-term psychological benefits for conditions such as depression and anxiety. However, these improvements tend to decline shortly after the cessation of the programs [19]. In contrast, pickleball has been recognized for its inclusive nature and appeal across different generations [20], making it particularly relevant for individuals with mental health disorders. Studies under review highlight the significant role and potential impact of pickleball in the mental health arena. On the one hand, it enhances personal well-being and life satisfaction, acting as a catalyst for positive health effects. On the other hand, pickleball serves as a preventative tool, helping to diminish symptoms of depression and showing substantial benefits in this respect among its participants. This aligns with the broader trend observed in other physical and sports activities, where an increased frequency of physical activity correlates with improved mental health [21,22]. This study hypothesized that a nine-week pickleball intervention would lead to significant improvements in self-esteem and reductions in psychiatric symptoms among individuals with schizophrenia.

## 2. Material and Methods

### 2.1. Participants

This research was approved by the Ethics Committee of Taichung Jen-Ai Hospital (No. 110-95). The study included a total of 30 participants, with 15 in the experimental group and 15 in the control group. The mean age of participants was 45.6 ± 7.8 years, ranging from 32 to 60 years. In terms of gender distribution, the experimental group consisted of 12 men and 3 women, while the control group consisted of 11 men and 4 women. Thirty schizophrenia participants were enlisted from long-term care institutions and a regional hospital (Figure 1). Prior to their admission into the study, local medical personnel assessed each participant for overall health and potential exercise contraindications. Criteria for exclusion included having sustained lower extremity injuries in the past year, experiencing any cardiac arrest incidents, or having diagnoses of neuromuscular disorders. We also omitted individuals who engaged in organized physical training exceeding two hours weekly, those who had participated in racquet sports within the last five years, or those with a background in competitive racquet sports. Moreover, participants who received medical advice to restrict physical activity were also precluded from the study. Demographic variables (age, gender, education, etc.) of each patient were collected by questionnaire and the electronic medical record.

### 2.2. Pickleball Intervention

Participants in the experimental group underwent a nine-week pickleball intervention, with three sessions per week, each lasting 90 min. Each session included specific components such as warm-up exercises, basic hitting technique training, dink shot practice, and cool-down exercises (Appendix A). The sessions were conducted under the guidance of a certified coach with a C-level license from the Taiwan Pickleball Association. Additionally, all sessions were jointly designed by the research team and occupational therapists from the institution, taking into account the physical and psychological characteristics of the participants to ensure the suitability and safety of the program.

### 2.3. Assessment of Self-Esteem and Psychiatric Symptoms

The Rosenberg Self-Esteem Scale (RSES) is a widely used self-report instrument for evaluating individual self-esteem. Created by sociologist Dr. Morris Rosenberg, the scale is a ten-item Likert scale that measures positive and negative feelings about the self. Each item is a statement related to self-worth or self-acceptance and is scored on a scale ranging from strongly agree to strongly disagree. The RSES is considered reliable and valid for assessing self-esteem in various groups and has been used in numerous research studies across different cultures and settings.

The Brief Symptom Rating Scale (BSRS-5) is a psychological self-report questionnaire designed to assess the severity of psychiatric symptoms in individuals, particularly focusing on those that are common in outpatient settings. It is a condensed version of the longer scales that are used to screen for a variety of symptoms across different psychiatric disorders [23].

### 2.4. Statistical Analysis

All statistical analyses were conducted using SPSS version 25.0. Descriptive statistics were applied to summarize baseline demographic and clinical characteristics, with data presented as frequencies, percentages, means, and standard deviations. The normality of the data was assessed using the Shapiro–Wilk test. Results indicated that the data for both the Rosenberg Self-Esteem Scale (RSES) and the Brief Symptom Rating Scale (BSRS-5) did not meet the normality assumption. Consequently, non-parametric statistical methods were used, including the Wilcoxon Signed-Rank Test for within-group comparisons and the Mann–Whitney U Test for between-group comparisons.

**Within-Group Analysis**: The Wilcoxon Signed-Rank Test was employed to analyze pre-test and post-test differences in the Rosenberg Self-Esteem Scale (RSES) and Brief Symptom Rating Scale (BSRS-5) within each group, assessing the significance of changes resulting from the intervention.

**Between-Group Analysis**: The Mann–Whitney U Test was used to compare the change scores (post-test minus pre-test) between the experimental and control groups, determining whether the pickleball intervention produced significant group-level differences.

**Supplementary Data Visualization**: Histogram charts were included to depict individual progress in the experimental group, particularly focusing on the Dink test results. These visualizations showcase weekly performance trends and illustrate improvements in motor skills and reaction times.

Statistical significance was set at *p* < 0.05, and all tests were two-tailed.

**Post Hoc Power Analysis**: A post hoc power analysis was performed to evaluate the sufficiency of the sample size for detecting medium effect sizes (d = 0.5). Results indicated a power of approximately 0.60, suggesting limited power for detecting smaller effects. Our study is exploratory in nature, and the sample size was primarily determined to provide foundational data that can serve as a reference for future large-scale studies. Future studies with larger sample sizes are recommended to enhance the robustness of the findings.

## 3. Results

Table 1 showed the demographic distribution of all participants. In terms of educational background, the majority had completed junior high school, with 12 participants (40.0%), followed by high school graduates, with 11 participants (36.7%). Regarding marital status, the largest group was single, with 25 participants (83.3%). There were 18 participants (60.0%) with no income, and 12 participants (40.0%) with an income of less than TWD 20,000. Since the participants were selected from a mental rehabilitation institution and most were in the recovery phase, it is common that many have limited or no income.

Table 2 and Table 3 present the within-group and between-group differences for the RSES in both the experimental and control groups before and after the pickleball intervention. The results indicate that there were no significant differences within either the experimental or control groups on the RSES scores before and after the pickleball intervention. Likewise, there were no significant differences between the two groups on the RSES after the intervention.

Table 4 and Table 5 show the within-group and between-group differences for the BSRS-5 for both the experimental and control groups before and after the pickleball intervention. The results reveal that there were no significant changes in the BSRS-5 scores within the experimental or control groups before and after the pickleball intervention. Furthermore, there were also no significant differences between the two groups on the BSRS-5 after the intervention.

Figure 2 showed the records of Dink ball tests conducted at the beginning, middle, and end of the training period, serving as an observation of the growth in the number of consecutive strikes by individual participants in the experimental group. The test data are presented in the form of histograms. During the nine-week pickleball intervention, there was a consistent improvement in the number of balls each participant could strike with the paddle. Particularly notable is the significant increase in the histogram shapes for all 15 participants when comparing data from before and after the intervention.

## 4. Discussion

This study aimed to investigate the effects of a nine-week pickleball intervention on self-esteem and psychiatric symptoms in individuals with schizophrenia. The findings of this study did not confirm the hypothesis that the pickleball intervention would lead to significant improvements in self-esteem or reductions in psychiatric symptoms among individuals with schizophrenia. While improvements in physical performance and attention were observed, the changes in self-esteem and psychiatric symptoms, as measured by the RSES and BSRS-5, were not statistically significant. This finding diverges from previous research that suggested exercise interventions could somewhat alleviate the positive and negative symptoms of schizophrenia. Our results do not align with the existing literature, which could be due to the level of physical activity or engagement in the sport without controls. Hornstrup and colleagues documented an observable boost in mental vigor (excluding enhancements in well-being or reductions in anxiety) among young female participants engaged in a handball regimen [24]. Conversely, Patterns et al. found that while an 8-week regimen of badminton or running heightened participants’ perceived and actual fitness, it did not markedly elevate their self-esteem compared to individuals who did not engage in any exercise [25]. Engagement in Karate-Do was associated with improvements in general mental health, emotional balance, and reductions in symptoms of depression and anxiety, in contrast to other activities (including physical exercises, cognitive tasks, and mindfulness practices) and a non-participating control group [26,27]. Ciaccioni et al. reported that a Judo regimen had no significant impact on the mental health or body image satisfaction of the participants [28], suggesting that the modest number of participants and the brief duration of the program may have influenced these outcomes. Such studies, which utilized literature reviews and meta-analyses, concluded that exercise could be an effective intervention for cognitive functions. However, it is important to note that these studies primarily focused on aerobic exercises and were of relatively short duration. Hence, while they were effective, the differences were not substantial [6,29,30]. Therefore, for pickleball exercises to potentially increase self-esteem in individuals with schizophrenia, it might be beneficial to design a longer duration of the exercise intervention and to adjust the intensity of the exercise sessions to achieve measurable outcomes.

We observed that pickleball intervention had no significant impact on symptom improvement in participants with schizophrenia, both before and after the course. This outcome is inconsistent with another study that employed Tai-Chi as an intervention, which showed a significant effect in treating both the positive and negative symptoms of schizophrenia [31]. The Tai-Chi intervention was part of a comprehensive treatment approach that, in addition to a year-long exercise program, included social skills training and maintained medication therapy. The sample size for that study was 244 participants, and the results indicated that Tai-Chi had a significant therapeutic effect on the negative symptoms of schizophrenia. By comparing that study to ours, it is apparent that our sample size is considerably smaller, and we need to consider increasing the number of participants. Furthermore, the shorter duration of the exercise intervention in our study could have affected the outcomes.

After a nine-week pickleball intervention, there was an enhancement in the participants’ reflexes and a significant improvement in their attention to the ball. This suggests that individuals dealing with extrapyramidal syndrome side effects from medication for schizophrenia have made some breakthroughs in their physical mobility challenges. Although the results of this study were not statistically significant, there was indeed a positive development in the exercise capabilities of the experimental group. This finding is similar to another study by Andersen et al., where 82 patients with schizophrenia underwent a high-intensity interval physical training program twice a week for 12 weeks. The study aimed to measure significant differences in the participants’ maximum oxygen uptake and cardiorespiratory fitness. No significant differences were found, but there was a slight improvement in oxygen uptake among all participants [32]. Hence, it is proposed that designing a more rigorous physical training regimen, coupled with professional medical advice and prescriptions, could potentially yield different outcomes.

This study has several limitations. First, the sample size is too small. Second, there are differences in the organizational management systems of the institutions recruiting participants; some participants could work outside during the day and return to the institution at night, while hospitals operated under a closed management system, which may confound the experimental results and data. Third, the cases of schizophrenia are inherently complex with positive and negative symptoms, and their physical and mental conditions are highly variable, making it difficult to control or maintain symptom stability. Fourth, gender may influence participation and intervention effects in pickleball activities, such as differences in physical abilities and psychological responses. These variables were not thoroughly analyzed in this study. Fifth, there was a lack of functional capacity assessment or heart rate analysis to verify the physical activity levels of the participants. Lastly, the duration of the experiment was only nine weeks. The potential of pickleball as an intervention for improving self-esteem and alleviating symptoms in schizophrenia is plausible. Future research will require more rigorous study designs and additional studies to confirm this possibility.

## 5. Conclusions

According to the design of this study, there were no significant differences observed in self-esteem enhancement or symptom alleviation in individuals with schizophrenia before and after the pickleball intervention. However, there was a notable increase in individual hit counts during the Dink ball tests, indicating that the pickleball intervention had a positive effect on the participants’ motor responsiveness and attention. Future studies and varied experimental designs are needed to further validate the impact of pickleball interventions on individuals with schizophrenia.

## Figures and Tables

**Figure 1 sports-13-00021-f001:**
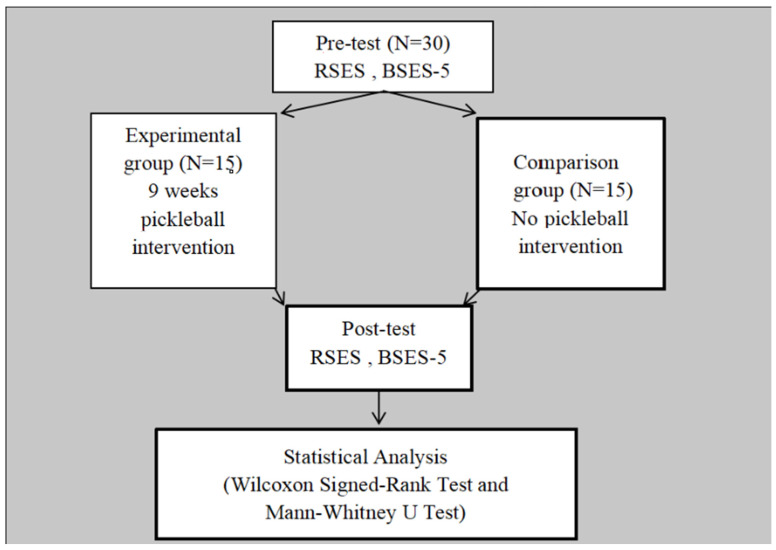
Flow chart of this study.

**Figure 2 sports-13-00021-f002:**
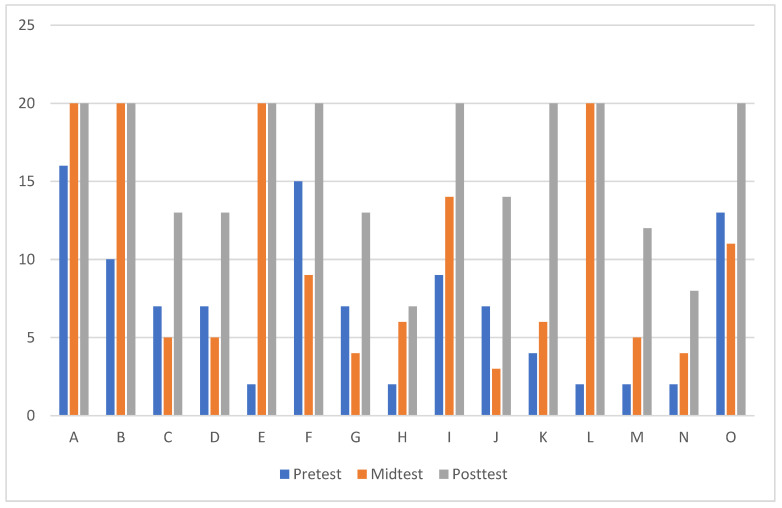
Individual hitting record histogram.

**Table 1 sports-13-00021-t001:** Demographic distribution of all participants.

Variable	Category	Number	Percentage	Experiment N = 15	Control N = 15
gender	male	16	53.30%	5	11
	female	14	46.60%	10	4
age	30–39	4	13.20%	4	0
	40–49	11	36.60%	7	4
	50–59	9	23.10%	2	7
	60–69	6	19.80%	2	4
education	elementary	1	3.30%	1	0
	junior	12	40.00%	2	10
	senior	11	36.70%	9	2
	university	5	16.70%	2	3
	graduate	1	3.30%	1	0
marriage status	single	25	83.30%	13	12
	married	1	3.30%	0	1
	divorce	4	13.30%	2	2
salary	none	18	60.00%	7	11
	Below NTD 20,000	12	40.00%	8	4

NTD: New Taiwan Dollars.

**Table 2 sports-13-00021-t002:** RSES differences within the experimental group and the control group.

	Time 0		Time 1		*p* Value
	Mean	SD	Mean	SD	
experimental	21.6	4.6	21.6	4.5	0.725
control	22.0	2.2	21.8	3.7	1.0

SD, standard deviation, Time 0, pre-test; Time 1, post-test.

**Table 3 sports-13-00021-t003:** RSES differences between the experimental group and the control group.

	Experimental		Control		*p* Value
	Mean	SD	Mean	SD	
RSES	−0.23	3.73	−0.20	4.88	0.783

SD, standard deviation.

**Table 4 sports-13-00021-t004:** BSRS-5 differences within the experimental group and the control group.

	Time 0		Time 1		*p* Value
	Mean	SD	Mean	SD	
experimental	9.67	2.77	8.86	3.62	0.36
control	9.73	5.04	10.27	4.65	0.75

SD, standard deviation, Time 0, pre-test; Time 1, post-test.

**Table 5 sports-13-00021-t005:** BSRS-5 differences between the experimental group and the control group.

	Experimental		Control		*p* Value
	Mean	SD	Mean	SD	
**BSRS-5**	−0.23	3.73	−0.20	4.88	0.44

SD, standard deviation.

## Data Availability

All the data will be available upon motivated request to the corresponding author of the present paper.

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
