# Peer review of "Effects of Pickleball Intervention on the Self-Esteem and Symptoms of Patients with Schizophrenia"

_sports, 2025, doi:10.3390/sports13010021_

Round 1
Reviewer 1 Report
Comments and Suggestions for Authors
The article addresses an important topic of using physical activity as a supportive intervention in the treatment of individuals with mental health disorders. However, its introduction requires significant expansion to fully justify the choice of the research method and enhance the scientific value of the work. While the authors highlight that regular physical activity positively impacts the mental, physical, and social health of individuals with schizophrenia, they do not provide detailed information about the mechanisms underlying these relationships. The introduction should include a literature review addressing the neurobiological and psychological benefits of physical exercise for individuals with mental disorders, specifically its effects on brain structure, emotion regulation, and cognitive functions.
Another critical omission is the lack of explanation regarding the choice of pickleball as an intervention for this particular group. This sport is relatively unfamiliar, and there is no initial justification for why it was deemed suitable in the context of schizophrenia, particularly considering its potential impact on motor skills, coordination, and social interactions. The authors could consider including an analysis of the literature on the effectiveness of similar team sports in treating mental disorders or highlighting the unique features of pickleball that may benefit this population.
The article does not present clearly defined research hypotheses or questions that could guide the analysis and interpretation of the findings.
Although the study’s methodology is briefly described, it requires more detailed development. Detailed information about the statistical analysis methods and the tests used to compare results between groups is lacking. It would also be advisable to apply more advanced data analysis methods to reveal subtle effects of the intervention.
A further point of critique is the limited and outdated bibliography. The article cites only 28 references, some of which are incomplete, and many of the sources are over a decade old. To strengthen the scientific credibility of the study, the authors should incorporate a larger number of references, ensuring that they include more recent publications. This would not only provide a more current context but also enhance the reliability of the theoretical foundations and the study’s findings.
Reviewer 2 Report
Comments and Suggestions for Authors
I’m more than glad for the opportunity to read and review the paper by Tsai-Chieh Chien and Chao-Chien Chen. The team presented a research approved by the ethical committee. I found this study interesting. However, I have some questions that must be clarified. Moreover, I also recommend some revisions to their protocol.
- I suggest shortening the conclusions of your abstract and enriching the results. Perhaps, provide exact values oof statistical tests, etc.
- The introduction is well-written in general. However, it would be welcome to see it stratified into smaller parts of the text.
- Have you ensured power analysis of your sample size?
- Did you follow EQUATOR guidelines? If yes, provide the appropriate checklist.
- Your methods are very shallow. This must be improved with several detailes provided.
- Your description of statistical analysis is too short, with a lot of information lacking.
- Provide surveys that you used in the supplementary file.
- There is a very unequal distribution of sex between study groups. It is a strong limitation.
- Start the discussion with the main findings of your research.
To sum up my review, this protocol needs deep revision. I suggest considering my comments during the revision process.
Comments on the Quality of English LanguageThe English is generally fine. However, this paper will benefit from proofreading.
Reviewer 3 Report
Comments and Suggestions for Authors
I would like to thank the authors for their article.
Below are my comments:
Comment 1: The authors should add a consort diagram showing the study design to the materials and methods section.
Comment 2: How was the sample size defined? Since a total of 30 participants equally divided into two groups is a small sample size.
Comment 3: How was schizophrenia defined? Which was the patient's pharmacological remedy?
Comment 4: The Pickleball Intervention section requires additional information, as it is too brief.
Comment 5: Did the authors evaluate the effect of exercise on the functional capacity of the exercise and control group participants? Investigation of possible correlations between functional capacity and the Rosenberg Self-Esteem Scale/Brief Symptom Rating Scale is highly important to support the study's novelty. The study cannot just use two scales as materials.
Comment 6: Based on comment 4, the results and discussion section needs to change drastically.
Comment 7: English must be improved.
Comments on the Quality of English LanguageEnglish must be improved
Reviewer 4 Report
Comments and Suggestions for Authors
Dear Authors,
Thank you for the opportunity to review your article. I found it intriguing to learn about a new sport, Pickleball, and its connection to the scientific area that I am passionate about: the relationship between physical activity and the mitigation or improvement of mental health symptoms. Your study is commendable for its effort to conduct an intervention study, which is relatively scarce in the literature. I appreciate that your manuscript contributes to this line of research. However, I have several suggestions that could help improve the article, as I believe there are several sections that require a thorough revision for it to be publishable. I hope you find my feedback helpful and consider the points that you deem pertinent.
Introduction
In my opinion, the introduction is too brief. The purpose of the introduction is to provide context for the entire investigation. While it acknowledges the variables, it only briefly touches on schizophrenia and the potential benefits of physical activity on mental health. This section needs further development. For instance, it should explain why Pickleball was chosen as the sport for this study. Although many diseases are mentioned, the focus is solely on schizophrenia. Additionally, the title suggests that the study examines the effects on both schizophrenia and self-esteem, yet the term "self-esteem" is not even mentioned in the introduction. It is crucial to define what self-esteem is, how it relates to sports, and its connection to schizophrenia. Furthermore, the rationale for conducting an intervention should be clarified. The literature review should distinguish between cross-sectional studies and intervention studies, explaining why an intervention was chosen. Lastly, the introduction should provide more details about Pickleball, such as its physical demands, cognitive components, and decision-making aspects.
Method
Participants
The section on participants should specify the final number of participants, the mean age, and the gender distribution (i.e., how many were men and how many were women).
Intervention and Variables
The intervention section lacks details that could affect the study's validity, such as the amount and intensity of physical activity. There is no analysis of heart rate or perceived exertion, making it difficult to verify the physical activity levels of the participants. It is important to control for these variables to ensure that all participants engaged in a comparable amount of physical activity.
Statistical Analysis
You mention using the Wilcoxon-Mann-Whitney test, which implies that a normality test was conducted and found non-significant. Please confirm this and include it in the manuscript, stating that a normality test was performed and the data did not meet the normality assumption, hence the use of non-parametric statistics.
Results
The results section needs improvement in terms of clarity and adherence to APA guidelines. Some information currently in the results section belongs in the methodology section. The results should be presented in accordance with the journal's guidelines and APA standards.
Discussion
The discussion should begin by restating the study's objectives and summarizing the observed results before delving into the interpretation of the findings. It should include the study's hypotheses, which are currently missing, and discuss the results in relation to these hypotheses. Some results do not align with the existing literature, which could be due to the level of physical activity or engagement in the sport. Without knowing if the physical activity level or engagement was controlled, it is difficult to draw definitive conclusions. Additionally, the discussion contains information that would be more appropriate in the introduction, such as the paragraph starting with "Various sports programs, like those evaluated."
Round 2
Reviewer 1 Report
Comments and Suggestions for Authors
The authors have responded positively to many of my comments, supplementing the text, for which I am grateful. However, I believe that the formulation of the research hypothesis established prior to the commencement of the study, as well as the research questions posed, should be included in the article at the end of the introduction. Subsequently, the discussion should address whether the hypothesis was confirmed and whether the research questions were answered. This is still missing in the article, and in my opinion, it is crucial for the potential acceptance of this article for publication.
Author Response
Reviewer: The authors have responded positively to many of my comments, supplementing the text, for which I am grateful. However, I believe that the formulation of the research hypothesis established prior to the commencement of the study, as well as the research questions posed, should be included in the article at the end of the introduction. Subsequently, the discussion should address whether the hypothesis was confirmed and whether the research questions were answered. This is still missing in the article, and in my opinion, it is crucial for the potential acceptance of this article for publication.
Authors: Thanks for your valuable comment,
We add “This study hypothesized that a nine-week pickleball intervention would lead to significant improvements in self-esteem and reductions in psychiatric symptoms among individuals with schizophrenia.” in the end of introduction.
We add “This study aimed to investigate the effects of a nine-week pickleball intervention on self-esteem and psychiatric symptoms in individuals with schizophrenia. The findings of this study did not confirm the hypothesis that the pickleball intervention would lead to significant improvements in self-esteem or reductions in psychiatric symptoms among individuals with schizophrenia. While improvements in physical performance and attention were observed, the changes in self-esteem and psychiatric symptoms, as measured by the RSES and BSRS-5, were not statistically significant.” in the beginning of discussion.
Reviewer 2 Report
Comments and Suggestions for Authors
The authors revised their paper properly.
Author Response
Thank you very much for your valuable comments.
Reviewer 3 Report
Comments and Suggestions for Authors
The authors successfully answer the recommended comments and revise their manuscript. I have no more comments to add.
Comments on the Quality of English LanguageEnglish is a little better now.
Author Response
Thanks for your valuable comments.
Reviewer 4 Report
Comments and Suggestions for Authors
Dear Authors,
Thank you for addressing my suggestions. I believe the article is now suitable for publication in the journal. Congratulations on your work and happy holidays.
Author Response
Thanks for your valuable comments.